# Towards Sustainable Investment Policies Informed by Opponent Shaping

**Juan Agustin Duque**\*, **Razvan Ciuca**\*, **Ayoub Echchahed**‡, **Hugo Larochelle**†, **Aaron Courville**†
University of Montreal and MILA
`firstname.lastname`@umontreal.ca
\*Equal contribution    †Equal supervision    ‡Core contributor

## Abstract

Addressing climate change requires global coordination, yet rational economic actors often prioritize immediate gains over collective welfare, resulting in social dilemmas. InvestESG is a recently proposed multi-agent simulation that captures the dynamic interplay between investors and companies under climate risk. We provide a formal characterization of the conditions under which InvestESG exhibits an intertemporal social dilemma, deriving theoretical thresholds at which individual incentives diverge from collective welfare. Building on this, we apply Advantage Alignment, a scalable opponent shaping algorithm shown to be effective in general-sum games, to influence agent learning in InvestESG. We offer theoretical insights into why Advantage Alignment systematically favors socially beneficial equilibria by biasing learning dynamics toward cooperative outcomes. Our results demonstrate that strategically shaping the learning processes of economic agents can result in better outcomes that could inform policy mechanisms to better align market incentives with long-term sustainability goals.

## 1 Introduction

Climate change is one of the defining challenges of our time, which demands immediate, coordinated global action to mitigate catastrophic environmental and economic consequences (Pörtner et al., 2022). Despite international agreements such as the Paris Agreement, nations continually grapple with aligning individual economic incentives with collective environmental goals. This misalignment can be effectively modeled as a *social dilemma*: a scenario in which players, acting independently and rationally according to their own interest, produce a suboptimal collective outcome (Rapoport and Chammah, 1965). In climate negotiations, nations often prefer short-term economic benefits over the collective welfare of aggressive mitigation strategies, leading to Pareto-suboptimal equilibria.

Previous advances in reinforcement learning (RL) have demonstrated remarkable capabilities in optimizing agent behaviors in various complex tasks (Mnih et al., 2013; Schulman et al., 2017). However, when confronted with general-sum games such as climate change negotiations, traditional RL methods consistently converge to selfish strategies that maximize immediate individual rewards but result in globally suboptimal outcomes (Sandholm and Crites, 1996; Foerster et al., 2018). These greedy equilibria do not realize the potential gains achievable through cooperation, underscoring the critical need for methods that inherently incentivize collaborative behavior among rational players.

To address this fundamental limitation, *opponent shaping* (Foerster et al., 2018) has emerged as a promising approach, allowing players to strategically influence the learning dynamics of other participants. Advantage Alignment, a recent advancement within opponent shaping algorithms (Duque et al., 2025), operates by aligning the *advantages* of interacting players. Unlike earlier opponent shaping methods that rely on computationally expensive imagined parameter updates (Foerster et al., 2018; Willi et al., 2022), Advantage Alignment modifies policy gradients directly, promoting actions that benefit both individual and collective outcomes efficiently.

Conventional MARL methods (IPPO (Yu et al., 2022)) optimize myopic returns, while recent opponent-shaping approaches (LOLA (Foerster et al., 2018), M-FOS (Lu et al., 2022), BRS Aghajohari et al. (2024a)) scale poorly beyond simple games or are limited to discrete action spaces

(LOQA (Aghajohari et al., 2024b)). In this paper, we apply Advantage Alignment in the context of sustainable investment policies in the InvestESG environment (Hou et al., 2025), a realistic simulation of corporate and investor interactions under climate risk. Specifically, our contributions are as follows.

- We formally prove for a simplified InvestESG that the parameter $\alpha$ (the climate responsiveness to mitigation) is critical for making the game a social dilemma in the stochastic setting (see Appendix D), and empirically show that the full game becomes a dilemma with $\alpha = 70$.
- We adapt Advantage Alignment to handle the complex dynamics characteristic of the original InvestESG simulation, without any simplifying assumptions, demonstrating its scalability and effectiveness compared to IPPO and MAPPO baselines (Yu et al., 2022).
- We provide a theoretical argument for why Advantage Alignment finds better equilibria than other baselines (including PPO with incentives and PPO summing the rewards).
- We provide insights into the policies discovered by Advantage Alignment, which contribute to mitigation more effectively than other algorithms.

## 2 BACKGROUND

### 2.1 MARKOV GAMES

We consider the formalism of $n$-player, fully observable, general-sum Markov Games (Shapley, 1953) which are represented by a tuple: $\mathcal{M} = (N, \mathcal{S}, \mathcal{A}, P, \mathcal{R}, \gamma)$. Here, the state space is denoted $\mathcal{S}$. The joint action space for all players in the game is written as $\mathcal{A} := \mathcal{A}^1 \times \ldots \times \mathcal{A}^n$. The transition function, $P : \mathcal{S} \times \mathcal{A} \to \Delta(\mathcal{S})$, maps from every state and joint action to a distribution over states. The reward structure is given by $\mathcal{R} = \{r^1, \ldots, r^n\}$, where each $r^i : \mathcal{S} \times \mathcal{A} \to \mathbb{R}$ specifies the reward received by player $i$. The discount factor, $\gamma \in [0, 1]$, determines the relative importance of future rewards.

### 2.2 REINFORCEMENT LEARNING

Consider $n$ players interacting in an environment with policies $\pi^i$, with $i \in [1, \ldots, n]$, each parameterized by $\theta^i$. Here, the result for each player is determined not only by their own actions but also by those of other players (von Neumann and Morgenstern, 1944). Following the convention of Agarwal et al. (2021), we can write the probability of a trajectory $\tau$ under the policies of all players as follows: $\mathrm{Pr}_\mu^{\pi^i, \pi^{-i}}(\tau) = \mu(s_0) \cdot \prod_{i=1}^n \pi^i(a_0^i | s_0) \cdot P(s_1 | s_0, \mathbf{a}_0) \ldots$ Where $\mu$ denotes the initial distribution over states, and $P(\cdot | s, \mathbf{a})$ is the transition distribution over states, which depends on the current state, $s$, and the joint action profile of all players, $\mathbf{a}$. In reinforcement learning the goal is to maximize the expected sum of future discounted reward, i.e. the value function:

$$V^i(\mu) \equiv \mathbb{E}_{\tau \sim \mathrm{Pr}_\mu^{\pi^i, \pi^{-i}}} \left[ R^i(\tau) \right] = \mathbb{E}_{\tau \sim \mathrm{Pr}_\mu^{\pi^i, \pi^{-i}}} \left[ \sum_{t=0}^\infty \gamma^t r^i(s_t, \mathbf{a}_t) \right]. \tag{1}$$

We also define the action-value function, also known as the $Q$-value, to define the advantage function:

$$Q^i(s, \mathbf{a}) = \mathbb{E}_{\tau \sim \mathrm{Pr}_\mu^{\pi^i, \pi^{-i}}} \left[ R^i(\tau) | s_0 = s, \mathbf{a}_0 = \mathbf{a} \right], \quad A^i(s, \mathbf{a}) = Q^i(s, \mathbf{a}) - V^i(s). \tag{2}$$

To optimize the value function, policy gradient methods perform gradient ascent on the policy parameters following a REINFORCE estimator (Williams, 1992) of the form:

$$\nabla_{\theta^i} V^i(\mu) = \mathbb{E}_{\tau \sim \mathrm{Pr}_\mu^{\pi^i, \pi^{-i}}} \left[ \sum_{t=0}^\infty \gamma^t A^i(s_t, \mathbf{a}_t) \nabla_{\theta^i} \log \pi^i(a_t^i | s_t) \right]. \tag{3}$$

The most popular variant of policy gradient algorithms, Proximal Policy Optimization (PPO) (Schulman et al., 2017), makes conservative updates to the policy using a clipping heuristic:

$$\mathbb{E}_{\tau \sim \mathrm{Pr}_\mu^{\pi^i, \pi^{-i}}} \left[ \min \left\{ \frac{\pi^i(a_t^i | s_t; \theta^i)}{\pi^i(a_t^i | s_t; \theta_l^i)} A^i(s_t, \mathbf{a}_t), \ \mathrm{clip}\left( \frac{\pi^i(a_t^i | s_t; \theta^i)}{\pi^i(a_t^i | s_t; \theta_l^i)}; 1 - \epsilon, 1 + \epsilon \right) A^i(s_t, \mathbf{a}_t) \right\} \right], \tag{4}$$

Where $\theta_l^i$ denotes the policy parameters of player $i$ after $l$ gradient updates. Intuitively, PPO attempts to keep the updated policy close to the original policy in terms of total variation distance.

## 2.3 SOCIAL DILEMMAS

Social dilemmas, as introduced by Rapoport and Chammah (1965), describe multi-player scenarios in which individual rational play leads to an outcome with less utility than that which would have occurred if all players had cooperated. Formally, consider a normal-form game with $n$ players, where each player $i$ selects an action $a^i$ from their action space $\mathcal{A}^i$, and receives utility $r^i(a^1, \ldots, a^n)$. Let $\mathbf{a}_C = (a_C^1, \ldots, a_C^n)$ denote a joint cooperative strategy profile that maximizes the total social welfare.

**Definition 1** (Nash Equilibrium (Nash, 1950)). *Consider an $n$-player game where each player $i \in \{1, \ldots, n\}$ has an action space $\mathcal{A}^i$ and a payoff function $r^i : \mathcal{A}^1 \times \cdots \times \mathcal{A}^n \to \mathbb{R}$. A strategy profile $\mathbf{a}_N = (a_N^1, \ldots, a_N^n)$ is a Nash equilibrium if for all players $i$:*

$$r^i(a_N^i, \mathbf{a}_N^{-i}) \geq r^i(a^i, \mathbf{a}_N^{-i}), \quad \forall a^i \in \mathcal{A}^i. \tag{5}$$

*Here, $\mathbf{a}_N^{-i}$ denotes the strategies chosen by all players other than player $i$.*

In this work we treat climate negotiations as a social dilemma. Informally, a repeated interaction is a social dilemma when there exists a cooperative pattern of behaviour that maximizes collective welfare, yet each individual actor has a short term incentive to deviate and free ride on the efforts of others. In such settings, myopic best responses tend to erode cooperation over time, even though all agents would be better off if they could sustain it. A formal definition of social dilemmas in Matrix games is given in Appendix A. For analyzing social dilemmas in Markov games, we define the *price of anarchy* (Nisan et al., 2007).

**Definition 2** (Price of Anarchy in Markov Games). *Let $\mathcal{M}$ be an $n$-player Markov game. Denote by $\Pi$ the set of joint policies and by $\mathcal{N} \subseteq \Pi$ the subset that are Nash (Definition 1). For a joint policy $\pi \in \Pi$ let $\mathcal{W}(\pi; \mu) := \sum_i V_\pi^i(\mu)$ be the social welfare function. The price of anarchy is*

$$\mathcal{P}_a = \frac{\max_{\pi \in \Pi} \mathcal{W}(\pi; \mu)}{\min_{\pi \in \mathcal{N}} \mathcal{W}(\pi; \mu)}. \tag{6}$$

A Markov game exhibits a social dilemma whenever $\mathcal{P}_a > 1$, i.e. the worst equilibrium achieves strictly lower total return than some feasible, $\mathcal{W}$-maximizing, joint policy. Intuitively, the price of anarchy is the ratio between the welfare of coordinated and individual rational play.

## 2.4 OPPONENT SHAPING

Opponent shaping, as introduced by Foerster et al. (2018), changes the paradigm of reinforcement learning: instead of assuming stationarity of the environment, opponent shaping algorithms assume that other players are learning agents. By making assumptions about the learning algorithms (Foerster et al., 2018) or modeling the problem as a meta-game (Lu et al., 2022), opponent shaping algorithms attempt to influence the learning dynamics of other players. Duque et al. (2025), proved that controlling other players via the Q-values is equivalent to doing policy gradient with the following modified advantage:

$$A^{*,i}(s_t, \mathbf{a}_t) = \left( A^i(s_t, \mathbf{a}_t) + \beta\gamma \cdot \sum_{j \neq i} \left( \sum_{k < t} \gamma^{t-k} A^i(s_k, \mathbf{a}_k) \right) A^j(s_t, \mathbf{a}_t) \right). \tag{7}$$

This Advantage Alignment formula can be plugged into the PPO formulation for a scalable Opponent Shaping algorithm (Proximal Advantage Alignment) that works in complicated multi-agent settings with high dimensional state representations, continuous action spaces, partial observability and multiple (more than 2) agents.

## 3 INVESTESG

InvestESG (Hou et al., 2025) is a multi-agent reinforcement learning (MARL) environment designed to study the long-term impact of corporate climate investments and ESG (Environmental, Social, and Governance) disclosure mandates in an intertemporal economy under climate risk. The environment

simulates interactions between company agents (who allocate capital across mitigation, resilience, and greenwashing strategies) and investor agents (who reallocate capital across companies based on profitability and ESG scores). Climate risk evolves over a 100-year horizon, shaped by cumulative mitigation efforts, and materializes through climate events that inflict economic losses on companies. By default, InvestESG uses 5 companies, 3 investors and doesn't allow for resilience or greenwashing strategies, although these parameters can be modified. The main limitations of this simulator, as a result of this parameterization, are clearly stated in the InvestESG paper (Hou et al., 2025).

### 3.1 ECONOMIC AND ENVIRONMENTAL DYNAMICS

For notational convenience, we use the index $i \in [1, \ldots, M]$ to refer to companies and index $j \in [1, \ldots, N]$ to refer to investors. In InvestESG, each investor's state is represented by their cash level $C_t^j$ and investment portfolio $\mathbf{S}_t^j = (H_{t,1}^j, \ldots, H_{t,M}^j)$, where $H_{t,i}^j$ denotes the investment that investor $j$ holds in company $i$ at timestep $t$. At each timestep, investors pick a binary portfolio vector $\mathbf{a}_t^j = [a_{t,1}^j, \ldots, a_{t,M}^j]$ representing which companies to invest in and companies choose to mitigate a percentage of its capital, $u_t^i \in [0, \bar{u}]$. The interim capital of each company at time $t + 1$ is given by:

$$K_{t+1,\text{interim}}^i = K_t^i - \underbrace{\sum_{j=1}^N H_{t,i}^j}_{\text{cash returned to investors}} + \underbrace{\sum_{j=1}^N a_{t,i}^j \frac{\mathcal{K}_t^j}{||\mathbf{a}_t^j||_1}}_{\text{new equity investments}}, \tag{8}$$

Where $K_t^i$ denotes the capital of company $i$ and $\mathcal{K}_t^j$ the capital of investor $j$ at time $t$. The reward at each timestep for company $i$ is given by the profit margin, $r_t^i = K_{t+1}^i - K_{t+1,\text{interim}}^i$. Climate dynamics, namely the risk that a particular type of harmful climate event occurs $P_t^e$ (where $e$ can be either an extreme heat, $h$, heavy precipitation, $p$, or drought, $d$, event) grows linearly w.r.t. initial event probability $P_0^e$ following:

$$P_t^e = \frac{\mu_e t}{1 + \lambda_e U_t} + P_0^e, \qquad \lambda_e = \alpha \times \tilde{\lambda}_e, \tag{9}$$

Where $\mu_e, \hat{\lambda}_e$, are calibration constants set to ensure that an annual mitigation investment of 2.3 trillion is required to achieve IPCC's 1.5°C scenario (Hou et al., 2025). We add an additional calibration constant, $\alpha$, that scales the $\hat{\lambda}$ parameters in order to make InvestESG a social dilemma (for details see section 4). The total risk is $P_t = 1 - (1 - P_t^h)(1 - P_t^p)(1 - P_t^d)$, and the cumulative mitigation $U_t$ is:

$$U_t = U_{t-1} + \sum_{i=1}^M u_t^i K_{t+1,\text{interim}}^i. \tag{10}$$

Finally, the capital of company $i$ is updated as follows:

$$K_{t+1}^i = (1 + \rho_t^i) K_{t+1,\text{interim}}^i, \quad \rho_t^i = (1 - u_t^i)(1 + \gamma)(1 - X_t L_i) - 1. \tag{11}$$

Where, $\rho_t^i$ is the profit margin of company $i$. Each type of harmful climate event (extreme heat, heavy precipitation or drought) is modeled independently by Bernoulli($P_t^e$), where $X_t \in \{0, 1, 2, 3\}$ is the total number of climate events realized at time $t$. $L_i$ above denotes the loss coefficient of company $i$ and $\gamma$ (which differs from the discount factor) is the market performance baseline.

## 4 A FORMAL ANALYSIS OF INVESTESG AS A SOCIAL DILEMMA

This section contains our main theoretical contribution: we show that, in InvestESG, the climate responsiveness parameter $\alpha$ is the critical quantity that determines whether the game behaves as a social dilemma, and we identify the corresponding threshold regime. Originally introduced as a social dilemma, the InvestESG environment is here studied in the stochastic setting with unseeded environmental dynamics, where we observe that the empirical price of anarchy (Definition 2) is close to 1 (Figure 5). The observed equilibrium does not change even when adding extremely strong ESG incentives such that investors effectively only care about ESG (the ESG=10 case in the figure). In a social dilemma, we would expect to see that adding prosocial incentives changes the final behavior of agents, yet this is not what we see empirically.

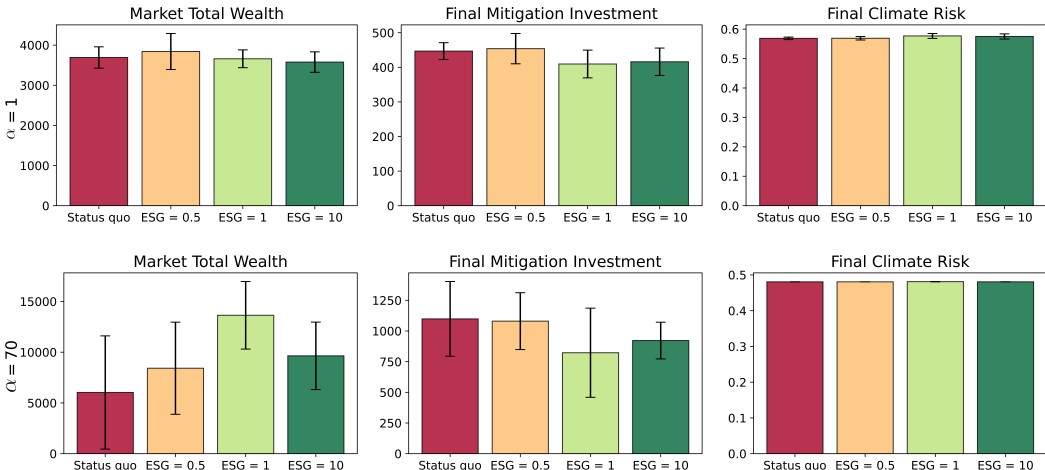

Figure 1: Comparison of final environment metrics at different $\alpha$ (introduced in equation 9) values, for 10 seeds of PPO agents: With the default scaling ($\alpha = 1$), the differences in market total wealth and final mitigation amount are negligible between agents with different ESG incentives. *Status Quo* refers to PPO agents trained without incentives. Increasing the scaling factor ($\alpha = 70$) results in higher market total wealth for ESG-conscious investors, and higher price of anarchy. The whiskers indicate a 1-standard deviation confidence interval. It should be noted that the ESG score is a multiplicative parameter, ESG = 10 represents an immense prosocial reward: the investors in that scenario effectively do not care at all about their own profits. These experiments were run with the full complexity allowed by InvestESG, without simplifications.

In this section, we derive conditions under which a simplified InvestESG explicitly meets the definition of a social dilemma, clarifying theoretically when this environment manifests the characteristic tensions between individual rationality and collective welfare. We then validate our theoretical results empirically in the full version of InvestESG without simplifications (See Figure 5). We find that the effectiveness of mitigation actions $\lambda$, namely how much each dollar invested in mitigation affects the climate event probability, is the critical parameter which makes InvestESG a social dilemma. We scale the mitigation effectiveness by multiplying the original parameter $\lambda$ by a scaling factor $\alpha$ (equation 9). Identifying this regime is essential, because it mirrors the real-world climate-mitigation problem in which near-term incentives myopically outweigh the long-term benefits of coordinated action for self-interested agents (Barrett, 1994). We look at three metrics: (1) **Market Total Wealth** is the combined value of all companies' capital and investors' cash at the end of the simulation; (2) **Final Mitigation Investment** is the total amount all companies have cumulatively spent on mitigation by that point; and (3) **Final Climate Risk** is the model-estimated probability that at least one damaging climate event occurs at the final timestep.

## 4.1 SET-UP AND NOTATION

Consider the amount of mitigation that company $i$ invests at time $t$, denoted $u_t^i$. Intuitively, a company cares about the marginal effect, or *benefit*, that this mitigation amount will have at the next time-step on their capital, $K_{t+1}^i$. Since $K_{t+1}^i$ is a random variable that depends on the realized number of climate events $X_t$, we take the marginal effect to be the partial derivative of the mitigation amount w.r.t. the expected value of the capital at the next time-step. See Appendix B for details.

**Definition 3** (Private Marginal Gradient). *Let $u_t^i$ be the mitigation amount of company $i$ at time $t$, then the private marginal gradient is*

$$\frac{d}{du_t^i}\mathbb{E}\left[K_{t+1}^i\right] = -\frac{\mathbb{E}[K_{t+1}^i]}{1 - u_t^i} + \mathbb{E}\left[\frac{(K_{t+1}^i)^2}{(1 - X_t L_i)^2(1 - u_t^i)(1 + \gamma)}\sum_e \frac{\lambda_e \mu_e t}{(1 + \lambda_e U_t)^2}\right] \quad (12)$$

*namely, the marginal effect in the capital of company $i$ by investing in mitigation at time-step $t$.*

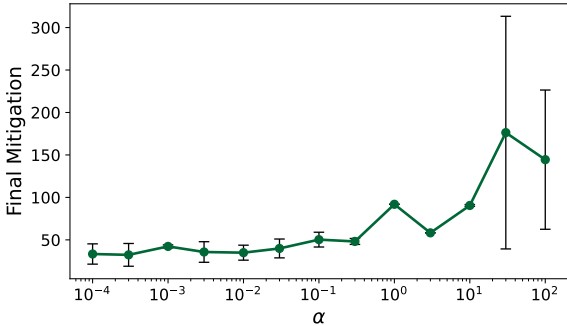

Figure 2: Final mitigation amount of 3 seeds of PPO agents trained in a single company and single investor environment for different values of $\alpha$. Here the public and private gradients are the same and we clearly see a threshold $\alpha \approx 30$ at which the final mitigation amount significantly increases. This threshold empirically marks the change of sign of the gradient. The whiskers indicate a 1-standard deviation confidence interval.

This gradient indicates whether a company has a selfish, myopic incentive to invest in mitigation: if the gradient is positive, then the capital at the next time-step increases for each dollar invested mitigating. Similarly, we define the social marginal gradient.

**Definition 4** (Social Marginal Gradient). *Let $u_t^i$ be the mitigation amount of company $i$ at time $t$, then the social marginal gradient is*

$$\frac{d}{du_t^i}\mathbb{E}\left[\sum_k K_{t+1}^k\right] = -\frac{\mathbb{E}[K_{t+1}^i]}{1 - u_t^i} + \sum_k \mathbb{E}\left[\frac{(K_{t+1}^k)^2}{(1 - X_t L_k)^2(1 - u_t^k)(1 + \gamma)} \sum_e \frac{\lambda_e \mu_e t}{(1 + \lambda_e U_t)^2}\right] \quad (13)$$

*namely, the marginal increase in the capital of all companies by company $i$ investing in mitigation at the next time-step.*

Our argument builds from the intuition that a social dilemma arises if there is a fundamental tension between these two gradients: the social marginal gradient pushes companies to mitigate, whereas the private gradient does not. The final proof, however, is more involved, as we analyze the effect of *all* previous mitigation actions on the final private and social capitals.

## 4.2 When is InvestESG a Social Dilemma?

The full stochastic dynamics of InvestESG are too complicated for exact analytic analysis so we simplify the environment. To isolate the *social-dilemma* component of climate-mitigation decisions, we work under the following assumptions for the remainder of this section.

**Assumption 1.** (Static Investor Portfolios). *Investor holdings are time–invariant. Formally, $H_{t,i}^j = H_{0,i}^j \,\forall\, t \geq 0,\ i \in [M],\ j \in [N]$. Thus investors never rebalance their capital allocations.*

**Assumption 2.** (Single-Company Allocation). *Each investor supplies capital to exactly one firm.*

**Assumption 3.** (No Historical Mitigation). *Prior to the period of interest, firms have undertaken no mitigation expenditure: $u_s^i = 0 \,\forall\, s < t,\ i \in [M]$.*

We are interested in computing the incentives on companies to start mitigating climate change in a world where no one has ever mitigated before and where the investor dynamics are simplified. These assumptions are not used in the experimental section. In this simplified world, we show that the existence of a social dilemma critically depends on the parameter $\lambda_e$ of the simulation, which is the responsiveness of climate change to mitigation investment. We give here an outline of the argument, the full derivation is in Appendix B:

1. There exists $\lambda_{\text{low}} > 0$ small enough such that in both the individual and social case, none of the derivatives $d\mathbb{E}[K_{t+1}^i]/du_{t-k}^i$ are positive.
2. In the individual case, there exists $\lambda$ such that at least some of the derivatives are positive, i.e. that it is possible to mitigate for self-interested reasons.

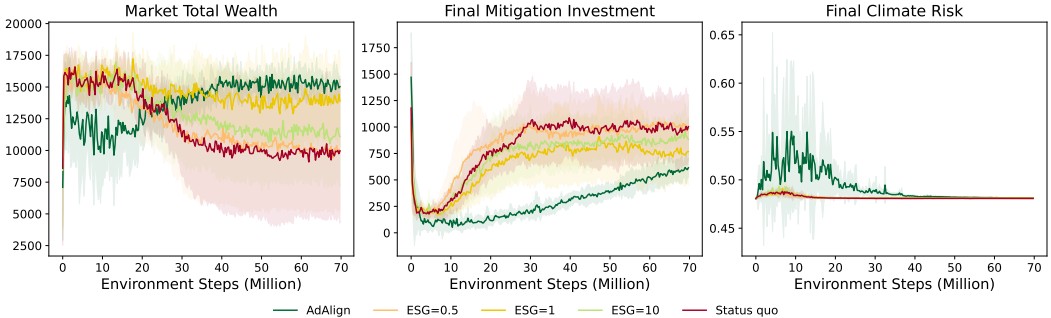

Figure 3: Training curves of PPO agents with different ESG values and Advantage Alignment (AdAlign) with $\alpha = 70$ for different metrics: Advantage Alignment agents achieve highest market total wealth by being more strategic about their mitigation investments compared to PPO agents. With significant lower final mitigation investment, AdAlign agents are able to achieve the same final climate risk and increase their capital returns. The shaded areas indicate a 1-standard deviation confidence interval. We note that $0.48$ is the best achievable climate risk in the environment, as it corresponds to $1 - \prod_e (1 - P_0^e)$, the floor of the probabilities of each event.

3. By intermediate-value theorem there is a $\lambda_{\text{critical}}$ where at least one of $d\mathbb{E}[K_{t+1}^i]/du_{t-k}^i$ is 0.

4. For the social case, the derivatives are *strictly* greater than those in the individual case i.e. company $i$'s mitigation efforts on all other companies is positive.

5. Therefore, there exists a $\lambda$ such that self-interested agents will not mitigate despite mitigation being beneficial for social welfare, thus we have a social dilemma.

This suffices to establish the existence of 3 zones in parameter space, defined by the two parameters $\lambda_{\text{low}}$ and $\lambda_{\text{critical}}$. For $\lambda < \lambda_{\text{low}}$, there is no dilemma because there is agreement between the signs of the individual and social gradients: mitigation is always net negative. If $\lambda_{\text{low}} \leq \lambda \leq \lambda_{\text{critical}}$, the gradient signs of the individual and social returns disagree, and we have a social dilemma. If $\lambda > \lambda_{\text{critical}}$, then there is again no strong dilemma and self-interested agents start mitigating myopically. For proofs and formal theorems for each of the steps in the outline see Appendix B. We empirically validate that this threshold exists by conducting an experiment in which we train three seeds of PPO agents, keeping the number of companies and investors fixed to one, but sweeping over the scaling parameter $\alpha$ (Figure 2). We find the threshold to be $\alpha = 70$, after running sweeping experiments with the default parameters (see Appendix E.1). When $\alpha$ increases beyond this threshold, PPO agents with different ESG incentives start to show different behaviors (Figure 5). We refer to this specific parameterization of InvestESG, with $\alpha = 70$, as $\alpha$-InvestESG.

# 5    APPLYING OPPONENT SHAPING TO INVESTESG

Having identified a regime in which InvestESG behaves as a social dilemma, we next model firms and investors as adaptive learners whose policies co-evolve in response to each other, reflecting that real market participants adjust their behavior over time rather than committing to a fixed equilibrium strategy. We apply Advantage Alignment (Duque et al., 2025) without ESG incentives to $\alpha$-InvestESG (found empirically in the previous section), in a centralized training decentralized execution manner (CDTE). Note that $\alpha$-InvestESG does not have the simplifying assumptions of the theoretical analysis and has the full complexity of the original simulator. Crucially, our Advantage Alignment implementation differs from the PPO implementation in two aspects. First, we substitute the advantage in the policy gradient with the Advantage Alignment expression (equation 7). Second, following the original paper we use *self-play*: one set of parameters for the company and another set of parameters for investor players. We found that self-play makes training more stable for Advantage Alignment (AdAlign) agents but hurts PPO agents, thus we compare the best version of each in Figure 3 (AdAlign with self-play, PPO without). Advantage Alignment agents outperform all PPO agents, even those with ESG incentives, while simultaneously having the lowest amount of final mitigation investment. For details about the policy parameterization and hyper-parameters used for these experiments refer to the Appendix D.

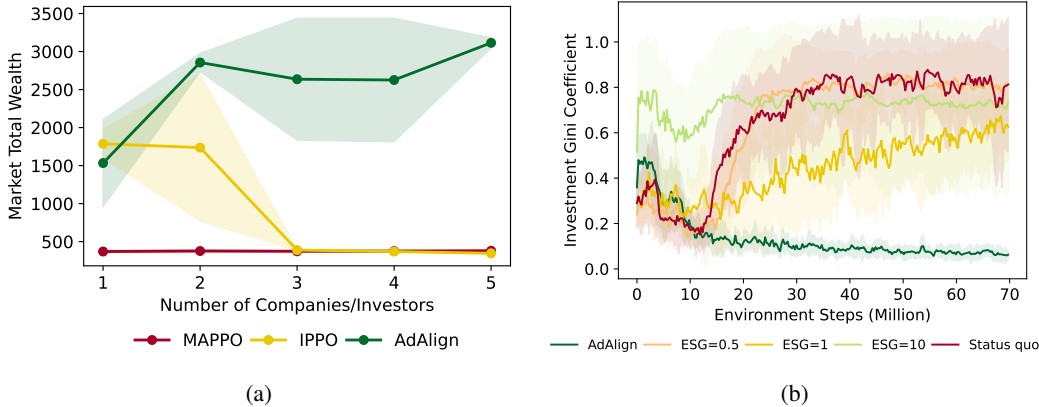

Figure 4: (a) Final market total wealth of 10 seeds of Advantage Alignment without ESG incentives, and PPO agents trained using summed rewards in InvestESG ($\alpha = 70$). On the x-axis we increase the number of companies and investors (1 company and 1 investor, 2 companies and 2 investors, etc.) while keeping the initial capital the same. Sum rewards is unable to find the action profile that maximizes social welfare once the number of players grows beyond a threshold ($> 2$), whereas Advantage Alignment consistently finds the same solution once the number of agents is large enough ($> 1$) going up to 10 agents. (b) Gini coefficient measuring inequality among company investments for different algorithms. Lower Gini indicates lower inequality and better distribution of resources. The shaded areas indicate a 1-standard deviation confidence interval.

## 5.1 Comparison with other Baselines

Intuitively if we care about social welfare, then we should try to maximize it explicitly. However, this does not yield good results in practice. We run experiments with summed rewards with the default InvestESG settings (5 companies, 3 investors), however the IPPO and MAPPO (Yu et al., 2022) agents always converged to sub-optimal equilibria. Our rationale for the baseline selection is discussed in the appendix D.3. Figure 4 (a) shows that naively summing the rewards scales poorly in the multi-agent setting. We run 10 seeds of IPPO and MAPPO agents that sum their rewards, scaling the number of agents in the environment, and plot the final market total wealth with initial fixed capital. Beyond 2 companies and 2 investors (4 agents total), PPO players summing the rewards are unable to find policies that maximize the social welfare. We hypothesize that the credit assignment problem becomes more difficult as the number of agents increases, giving noisy temporal difference signals to centralized critic methods like MAPPO. In contrast, Advantage Alignment maintains high social welfare across all experiments. We analyze the reasons why this happens in section 6.

## 5.2 Interpreting Advantage Alignment Policies

To assess how opponent shaping alters investment behaviors in the social-dilemma setting, we run simulations of Advantage Alignment (AA) and the standard PPO baseline with ESG score = 0, which means that there is no external incentive to mitigate. The policies obtained are summarized in Fig. 7. Three patterns emerge. (i) AA learns to mitigate just enough and only at the critical moments when climate risk increases, whereas PPO invests excessively. (ii) The capital allocations show that AA maintains an approximately uniform investment distribution across companies, while PPO concentrates wealth, replicating the disparities reported by Hou et al. (2025) (Figure 4(b)). (iii) AA agents learn to coordinate themselves to share mitigation costs, in contrast to PPO agents whose independent updates yield uncoordinated, non-cooperative mitigation investments.

## 6 On the Effectiveness of Advantage Alignment

Advantage Alignment is particularly effective at finding social welfare maximizing strategies in social dilemmas at least in part because it has a strong initial bias towards cooperative strategies. When advantages are estimated with Generalized Advantage Estimation (GAE; Schulman et al., 2018), Advantage Alignment nudges agents toward cooperation. For agent $i$ the transformed advantage is

$$A_t^{*,i} = A_t^i + \beta\gamma \sum_{j \neq i} \Big( \sum_{k < t} \gamma^{t-k} A_k^i \Big) A_t^j, \qquad (14)$$

with $A_t^i := A^i(s_t, \mathbf{a}_t)$. Define the on-policy mean $b^i := \mathbb{E}\left[\sum_{k<t} \gamma^{t-k} A_k^i\right]$. Decomposing the expectation gives

$$A_t^{*,i} = \underbrace{A_t^i + \beta\gamma b^i \sum_{j\neq i} A_t^j}_{\text{cooperative bias}} + \beta\gamma \sum_{j\neq i} \underbrace{\left(\sum_{k<t} \gamma^{t-k} A_k^i - b^i\right)}_{\text{zero-mean}} A_t^j. \tag{15}$$

The first term favors joint gains; if $\beta\gamma b^i = 1$, the cooperative term is exactly that encountered in summed-reward learning. With perfect critics, $b^i = 0$, this bias vanishes. However, during actor-critic training, the critic lags the improving policy, so the one-step TD error

$$\delta_t = r_t + \gamma V_\phi(s_{t+1}) - V_\phi(s_t), \tag{16}$$

is positively biased (see Kearns and Singh (2000)) for a formal argument based on contraction mappings): $\mathbb{E}[\delta_t] > 0 \implies \mathbb{E}[A_t^{\text{GAE}}] > 0 \implies b^i > 0$. Hence early training amplifies cooperative updates. As the critic catches up and TD errors become unbiased, $b^i \to 0$ and the cooperative bias fades. This initial bias prevents the algorithm from falling into defective equilibria early in training.

## 7 RELATED WORK

Opponent shaping treats co-learners as adaptive parts of the environment and deliberately steers their updates. LOLA (Foerster et al., 2018) introduced the idea by differentiating through a single naïve policy-gradient step of other players. Follow-up methods refined the update rule (Letcher et al., 2021; Zhao et al., 2022; Willi et al., 2022) or approximated best-response dynamics (Aghajohari et al., 2024a; Lu et al., 2022). LOQA replaced higher-order differentiation with REINFORCE-based control of opponents' Q-values, enabling scalability to larger games (Aghajohari et al., 2024b). Previous work on *Advantage Alignment* showed that many of these algorithms implicitly multiply agents' advantages and offered a simpler, computationally cheaper formulation that preserves Nash equilibria in general-sum games (Duque et al., 2025). The present paper leverages that insight to shape investor–company interactions in a realistic climate-finance simulator. A growing line of research embeds MARL agents inside climate models to study emergent behavior, test policy instruments, or optimize control strategies. RICE-N extends Nordhaus' regional integrated assessment model into a multi-agent game in which learning agents negotiate emission targets and investment pathways (Zhang et al., 2022). The *AI for Global Climate Cooperation* competition has since used this platform to study how AI negotiators can sustain long-run cooperation under enforcement uncertainty. Our work differs in scope: rather than modeling inter-government bargaining, we focus on the micro-level interaction between profit-seeking firms and ESG-motivated investors captured by InvestESG (Hou et al., 2025). Inspired by the *AI Economist* (Zheng et al., 2021), Wang et al. simulate a cap-and-trade market with enterprise and government agents; the government agent learns dynamic permit-allocation rules that balance output, equity, and emissions (Wang et al., 2024). Our study is complementary: we assume no central planner and instead demonstrate that shaping firm–investor learning alone can steer the economy toward socially desirable equilibria.

## 8 CONCLUSION

In this paper, we presented a rigorous theoretical and empirical examination of economic simulations under climate risk, using game theory. By formally characterizing the InvestESG environment as an intertemporal social dilemma, we established the precise conditions under which individual incentives diverge from collective sustainability goals. This allowed us to fine-tune InvestESG to capture the social dilemma conditions observed in the real world. Leveraging the scalable power of Advantage Alignment, we demonstrated that strategically shaping agent learning dynamics can effectively mitigate selfish incentives, systematically driving the equilibrium toward socially beneficial outcomes, even in the absence of government mandates. Our empirical results underscore the practical efficacy of opponent-shaping algorithms in complex, high-dimensional economic scenarios, addressing a critical gap where previous multi-agent RL methods have failed. By elucidating why Advantage Alignment naturally improves the credit assignment problem and finds more cooperative equilibria, we provided key insights that pave the way for robust, policy-relevant solutions capable of maximizing the social welfare more effectively. Beyond theoretical advancements, this work signifies a tangible step toward harnessing artificial intelligence to tackle real-world climate policy challenges and inform government decision-making.

## REPRODUCIBILITY STATEMENT

We provide all details needed to replicate our results. The full set of training hyperparameters, model architectures, optimizer choices, rollout lengths, discounting/GAE settings, clipping, and gradient/entropy coefficients for all methods (IPPO and Advantage Alignment) are listed in Appendix D, Tables 1 and 2. We also report the number of parallel environments, episode length, total environment steps, and (where applicable) the use of self-play, as well as the number of random seeds for each experiment in the figure captions. Upon de-anonymization, we will publicly release the complete codebase, including experiment configs, training scripts, and plotting utilities to reproduce all figures and tables from scratch.

## AKNOWLEDGMENTS

Juan Agustin Duque is supported by the St-Pierre-Larochelle Scholarship at the University of Montreal and by Aaron Courville's CIFAR AI Chair in Representations that Generalize Systematically.

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

APPENDIX

## A    SOCIAL DILEMMA DEFINITION

**Definition 5** (Social Dilemma). *Let $\mathbf{a}_N$ be a Nash equilibrium of the game. A game is said to be a social dilemma if it satisfies both of the following conditions:*

1. Collective Inefficiency: *The socially optimal outcome yields strictly higher total utility than the Nash equilibrium:*

$$\sum_{i=1}^{n} r^i(\mathbf{a}_C) > \sum_{i=1}^{n} r^i(\mathbf{a}_N). \tag{17}$$

2. Individual Incentive to Defect: *For each player $i$, there exists some joint strategy $\mathbf{a}^{-i}$ such that deviating from the cooperative strategy is individually beneficial:*

$$r^i(a_D^i, \mathbf{a}_C^{-i}) > r^i(a_C^i, \mathbf{a}_C^{-i}), \tag{18}$$

*where $a_D^i$ is a defection action and $\mathbf{a}_C^{-i}$ denotes the cooperative actions of all other players.*

These two conditions capture the tension at the heart of social dilemmas: cooperation is globally optimal, but locally unstable due to unilateral incentives to defect (Capraro and Halpern, 2015). However, this definition makes strong assumptions that may not hold in the real world (e.g. uniqueness of Nash).

## B  MATHEMATICAL DERIVATIONS

In this section we prove the main result of this work, namely, that InvestESG is a social dilemma for the correct choice of the climate mitigation effectiveness parameter $\lambda$. We begin by stating our assumptions about the environment:

**Assumption 1.** (Static Investor Portfolios). *Investor holdings are time–invariant. Formally, $H_{t,i}^j = H_{0,i}^j \, \forall \, t \geq 0, \ i \in [M], \ j \in [N]$. Thus investors never rebalance their capital allocations.*

**Assumption 2.** (Single-Company Allocation). *Each investor supplies capital to exactly one firm.*

**Assumption 3.** (No Historical Mitigation). *Prior to the period of interest, firms have undertaken no mitigation expenditure: $u_s^i = 0 \, \forall \, s < t, \ i \in [M]$.*

### B.1  INDIVIDUAL CASE ANALYSIS

The capital of a company is:

$$K_{t+1}^i = (1 + \rho_t^i) K_{t+1,\text{interim}}^i. \tag{19}$$

$$\rho_t^i = (1 - u_t^i)(1 + \gamma)(1 - X_t L_i) - 1. \tag{20}$$

$$K_{t+1,\text{interim}}^i = K_t^i - \sum_{j=1}^N H_{t,i}^j + \sum_{j=1}^N a_{t,i}^j \frac{\mathcal{K}_t^j}{||\mathbf{a}_t^j||_1}. \tag{21}$$

where $\mathcal{K}_t^j$ is the capital of the $j$-th investor. In this argument we will isolate the companies' behavior by assuming that the investors do not change their capital allocations over time, and that each investor invests in a single company. This means that the money returned to investors is the same as the money investors put in at the current time-step. Giving us

$$K_{t+1,\text{interim}}^i = K_t^i, \tag{22}$$

Plugging this into the equation for $K_{t+1}^i$ gives us:

$$K_{t+1}^i = \left( (1 - u_t^i)(1 + \gamma)(1 - X_t L_i) \right) K_t^i. \tag{23}$$

Here $X_t$ is the number of climate events at time $t$, it can go up to 3, if extreme heat, precipitation and drought all happen. The probability of each of those events in $X_t$ depends on the mitigation up to time $t$, called $U_t$:

$$P_t^e = \frac{\mu_e t}{1 + \lambda_e U_t} + P_0^e, \qquad \lambda^e = \alpha \times \tilde{\lambda}^e, \tag{24}$$

Let's consider a company which has never done any mitigation in the past, so $U_{t-1} = 0$, but is considering doing so at this time-step, we can differentiate $\mathbb{E}[K_{t+1}^i]$ with respect to $u_t^i$ to investigate if it has an incentive to start mitigating. If this derivative is positive, then the company has an immediate, myopic incentive to take mitigation actions.

**Lemma 1.** *For every time-step $t$, company $i$ and scalar $\alpha_i > 0$, we have that the social marginal gradient is strictly greater than the private marginal gradient.*

$$\frac{d}{du_t^i} \sum_j \alpha_j \mathbb{E}\left[ K_{t+1}^j \right] > \alpha_i \frac{d}{du_t^i} \mathbb{E}\left[ K_{t+1}^i \right]. \tag{25}$$

*Proof.*

$$\frac{d}{du_t^i} \mathbb{E}\left[ K_{t+1}^i \right] = \frac{d}{du_t^i} \mathbb{E}\left[ \left( (1 - u_t^i)(1 + \gamma)(1 - X_t L_i) \right) K_t^i \right]. \tag{26}$$

The only parts of this equation that depend on $u_t^i$ are the factor of $(1 - u_t^i)$ and through $(1 - X_t L_i)$. Now for small values of climate risk events, we can approximate $X_t$ as a Bernoulli variable with

probability $P_t = \sum_e P_t^e$, hence the expectation above can be written as a sum over $X_t = \{0, 1\}$, the algebraic trick used here comes from $f(x, y, z) = xyz \implies df/dx = yz = f(x, y, z)/x$, giving us:

$$\frac{d}{du_t^i}\mathbb{E}\left[K_{t+1}^i\right] = -\frac{\mathbb{E}[K_{t+1}^i]}{1 - u_t^i} - \mathbb{E}\left[\frac{K_{t+1}^i}{1 - X_t L_i}\frac{dP_t}{du_t^i}\right]. \tag{27}$$

And now for $\frac{dP_t}{du_t^i}$, referring back to equation 24:

$$\frac{dP_t}{du_t^i} = \frac{dU_t}{du_t^i}\frac{d}{dU_t}\sum_e\left(\frac{\mu_e t}{1 + \lambda_e U_t} + P_0^e\right) = K_{t+1,\text{interim}}^i\sum_e\frac{-\lambda_e\mu_e t}{(1 + \lambda_e U_t)^2}, \tag{28}$$

Where the factor of $K_{t+1,\text{interim}}^i$ comes from the fact that $U_t \equiv \sum_{s<t} u_s^i K_{s+1,\text{interim}}^i$. And finally:

$$\frac{d}{du_t^i}\mathbb{E}\left[K_{t+1}^i\right] = -\frac{\mathbb{E}[K_{t+1}^i]}{1 - u_t^i} + \mathbb{E}\left[\frac{K_{t+1}^i}{1 - X_t L_i}K_{t+1,\text{interim}}^i\sum_e\frac{\lambda_e\mu_e t}{(1 + \lambda_e U_t)^2}\right]. \tag{29}$$

This equation places a lower bound on how incentivized a selfish company is to invest in mitigation, it does not include the long-term effects of mitigation at all. If $\lambda_e$ is set high enough that this derivative is positive, we do not have a social dilemma at all. Noticing that $K_{t+1,\text{interim}}^i = K_{t+1}^i/(1 + \rho_t^i)$ (from the equations at the beginning of the section), we can replace:

$$\frac{d}{du_t^i}\mathbb{E}\left[K_{t+1}^i\right] = -\frac{\mathbb{E}[K_{t+1}^i]}{1 - u_t^i} + \mathbb{E}\left[\frac{(K_{t+1}^i)^2}{(1 - X_t L_i)^2(1 - u_t^i)(1 + \gamma)}\sum_e\frac{\lambda_e\mu_e t}{(1 + \lambda_e U_t)^2}\right]. \tag{30}$$

We can also trivially run the same steps for $\frac{d}{du_t^i}\mathbb{E}\left[K_{t+1}^j\right]$ with $j \neq i$, i.e. computing the derivative of the return of agent $j$ w.r.t. the mitigation of agent $i$, which will remove the first term in the previous equation. The cumulative mitigation $U_t$ includes terms from all agents, so the contribution of the $(1 - X_t L_j)$ term to the gradient remains:

$$\text{for } i \neq j : \frac{d}{du_t^i}\mathbb{E}\left[K_{t+1}^j\right] = \mathbb{E}\left[\frac{K_{t+1}^j K_{t+1}^i}{(1 - X_t L_j)(1 - X_t L_i)(1 - u_t^i)(1 + \gamma)}\sum_e\frac{\lambda_e\mu_e t}{(1 + \lambda_e U_t)^2}\right]. \tag{31}$$

We notice that this is strictly non-negative, as expected (since all $K_t^i$ are non-negative). All agents would want someone else to mitigate for them, they get all the benefits of mitigation without any of the costs. Together with the non-negativity of $K_t^i$, we get the following inequality:

$$\forall i \quad \frac{d}{du_t^i}\sum_j\mathbb{E}\left[K_{t+1}^j\right] > \frac{d}{du_t^i}\mathbb{E}\left[K_{t+1}^i\right], \tag{32}$$

which falls straight out of equation 31 being strictly non-negative. This means that the "social derivative" is always greater than the selfish derivative for mitigation actions $u_t^i$. In particular, we also get the extra inequality:

$$\forall t \forall i \in \{0, ..., n\}, \forall \alpha_i > 0 \quad \frac{d}{du_t^i}\sum_j\alpha_j\mathbb{E}\left[K_{t+1}^j\right] > \alpha_i\frac{d}{du_t^i}\mathbb{E}\left[K_{t+1}^i\right]. \tag{33}$$

which we will need later. $\square$

## B.2 DEPENDENCE ON ALL PREVIOUS MITIGATION ACTIONS

**Lemma 2.** *For every **previous** time-step $t - k$, $0 < k < t$, company $i$ and scalar $\alpha_i > 0$, we have that the social gradient is strictly greater than the private gradient.*

$$\frac{d}{du_{t-k}^i}\sum_l\alpha_l\mathbb{E}\left[K_{t+1}^l\right] > \frac{d}{du_{t-k}^i}\alpha_i\mathbb{E}\left[K_{t+1}^i\right]. \tag{34}$$

*Proof.* Writing the main equation again for visibility:

$$K_{t+1}^i = \left((1 - u_t^i)(1 + \gamma)(1 - X_t L_i)\right)K_t^i. \tag{23}$$

To compute the dependence of $K_{t+1}^i$ on $u_{t-k}^i$ for a general $k \geq 1$, we note that this enters in two ways in the equation above, first through the effect on $P_i$, which affects $K_{t+1}^j$ through $(1 - u_t^i)(1 + \gamma)(1 - X_t L_i)$, and this will give us a factor like the second one in equation 29. The second way is through effects on $K_t^i$, which were not present in the previous section:

$$\frac{d}{du_{t-k}^i}\mathbb{E}\left[K_{t+1}^i\right] = -\mathbb{E}\left[\frac{K_{t+1}^i}{1 - X_t L_i}\frac{dP_t}{du_{t-k}^i}\right] + \left((1 - u_t^i)(1 + \gamma)(1 - P_t L_i)\right)\frac{d}{du_{t-k}^i}\mathbb{E}\left[K_t^i\right]. \quad (35)$$

We have used the independence of $X_t$ and $K_t$ to push in the expectation values. These two variables are independent if the past mitigation actions are kept constant, which is the case for us, since we are analyzing the case where all companies have never invested in mitigation (by construction). $dP_t/du_{t-k}$ is computed as before:

$$\frac{dP_t}{du_{t-k}^i} = \frac{dU_t}{du_{t-k}^i}\frac{d}{dU_t}\sum_e\left(\frac{\mu_e t}{1 + \lambda_e U_t} + P_0^e\right) = K_{t+1-k,\text{interim}}^i\sum_e\frac{-\lambda_e\mu_e t}{(1 + \lambda_e U_t)^2}. \quad (36)$$

We can now use equation 35 to induce a recurrence relation between the derivatives of all factors $\frac{d}{du_{t-k}^i}\mathbb{E}\left[K_{t+1}^i\right]$. All put together, the recurrence relation is

$$\frac{d}{du_{t-k}^i}\mathbb{E}\left[K_{t+1}^i\right] = \mathbb{E}\left[\frac{(K_{t+1}^i)^2}{(1 - X_t L_i)^2(1 - u_t^i)(1 + \gamma)}\sum_e\frac{\lambda_e\mu_e t}{(1 + \lambda_e U_t)^2}\right]$$
$$+ \left((1 - u_t^i)(1 + \gamma)(1 - P_t L_i)\right)\frac{d}{du_{t-k}^i}\mathbb{E}\left[K_t^i\right]. \quad (37)$$

And for the derivative with a different agent $l \neq i$:

$$\frac{d}{du_{t-k}^i}\mathbb{E}\left[K_{t+1}^l\right] = \mathbb{E}\left[\frac{K_{t+1}^i K_{t+1}^l}{(1 - X_t L_i)(1 - X_t L_l)(1 - u_t^i)(1 + \gamma)}\sum_e\frac{\lambda_e\mu_e t}{(1 + \lambda_e U_t)^2}\right]$$
$$+ \left((1 - u_t^l)(1 + \gamma)(1 - P_t L_l)\right)\frac{d}{du_{t-k}^i}\mathbb{E}\left[K_t^l\right]. \quad (38)$$

We now wish to get the equivalent of lemma 1 for mitigations $u_{t-k}^i$ at any previous time-step. We use a proof by induction on the index $k$ from the base case of $k = 0$ in equation 33. Let $\alpha_i > 0$ be strictly positive real numbers:

$$\frac{d}{du_{t-k}^i}\sum_l\alpha_l\mathbb{E}\left[K_{t+1}^l\right] =$$

$$\sum_l\alpha_l\mathbb{E}\left[\frac{K_{t+1}^i K_{t+1}^l}{(1 - X_t L_i)(1 - X_t L_l)(1 - u_t^i)(1 + \gamma)}\sum_e\frac{\lambda_e\mu_e t}{(1 + \lambda_e U_t)^2}\right]$$
$$+ \sum_l\alpha_l\left((1 - u_t^l)(1 + \gamma)(1 - P_t L_l)\right)\frac{d}{du_{t-k}^i}\mathbb{E}\left[K_t^l\right]. \quad (39)$$

We now invoke the induction assumption for the term in the last line, which matches the induction case with $\alpha_l' = \alpha_l\left((1 - u_t^l)(1 + \gamma)(1 - P_t L_l)\right)$, this is why we needed the assumption with a positive linear combination. The assumption gives us the following:

$$\sum_l\alpha_l\left((1 - u_t^l)(1 + \gamma)(1 - P_t L_l)\right)\frac{d}{du_{t-k}^i}\mathbb{E}\left[K_t^l\right] >$$
$$\alpha_i\left((1 - u_t^i)(1 + \gamma)(1 - P_t L_i)\right)\frac{d}{du_{t-k}^i}\mathbb{E}\left[K_t^i\right], \quad (40)$$

But now we note that the elements in the first sum above are all positive, and so

$$\sum_l\alpha_l\mathbb{E}\left[\frac{K_{t+1}^i K_{t+1}^l}{(1 - X_t L_i)(1 - X_t L_l)(1 - u_t^i)(1 + \gamma)}\sum_e\frac{\lambda_e\mu_e t}{(1 + \lambda_e U_t)^2}\right] >$$
$$\alpha_i\mathbb{E}\left[\frac{(K_{t+1}^i)^2}{(1 - X_t L_i)^2(1 - u_t^i)(1 + \gamma)}\sum_e\frac{\lambda_e\mu_e t}{(1 + \lambda_e U_t)^2}\right], \quad (41)$$

This suffices to establish the desired inequality

$$\frac{d}{du_{t-k}^i} \sum_l \alpha_l \mathbb{E}\left[K_{t+1}^l\right] > \frac{d}{du_{t-k}^i} \alpha_i \mathbb{E}\left[K_{t+1}^i\right], \tag{42}$$

Which as a special case ($\alpha_l = 1$ for all $l$) tells us that the gradient of the social welfare with respect to the mitigation actions of any agent at any point in the past is higher than the selfish gradient. $\square$

### B.3 INDUCTION ARGUMENT RE-STATED

**Theorem 1** (Social-mitigation). *For all $\lambda > 0$, $\forall t, i, k$,*

$$\frac{d}{du_{t-k}^i} \mathbb{E}\left[\sum_j K_{t+1}^j\right] > \frac{d}{du_{t-k}^i} \mathbb{E}\left[K_{t+1}^i\right]. \tag{43}$$

*Proof.* In order to prove theorem 1, we need to prove the following equation for all $i, t, k$:

$$\frac{d}{du_{t-k}^i} \sum_l \alpha_l \mathbb{E}\left[K_{t+1}^l\right] > \frac{d}{du_{t-k}^i} \alpha_i \mathbb{E}\left[K_{t+1}^i\right], \tag{44}$$

we proceed by induction on $k$, using $k = 0$ as base case:

**Induction base case $k = 0$.** The base case is trivial from lemma 1.

**Induction recursion.** For the induction, we assume that equation 44 holds for all $k' < k$, and show that this implies that the equation is true for $k$. This is what equations 40 through 41 are showing.

By setting $\alpha_l = 1$ in the equation, we finish the proof. $\square$

### B.4 ALL SELFISH GRADIENTS ARE NEGATIVE AT LOW ENOUGH $\lambda$

**Theorem 2** (Low mitigation effectiveness). *There exists $\lambda_{low} > 0$ such that for all $\lambda < \lambda_{low}$, we have:*

$$\frac{d}{du_{t-k}^i} \mathbb{E}\left[K_{t+1}^i\right] < 0 \quad \text{and} \quad \frac{d}{du_{t-k}^i} \sum_j \mathbb{E}\left[K_{t+1}^j\right] < 0 \quad \forall i, t, k. \tag{45}$$

*Proof.* We proceed by simple recurrence to show that all gradients are negative at $\lambda_e = 0$. We note that equation 30 at $\lambda_e = 0$ reduces to

$$\frac{d}{du_t^i} \mathbb{E}\left[K_{t+1}^i\right] = -\frac{\mathbb{E}[K_{t+1}^i]}{1 - u_t^i}, \tag{46}$$

thus this gradient is negative. Using equation 37, we see that the first term vanished in the $\lambda_e = 0$, giving us a term which only depends on $\frac{d}{du_{t-k}^i}\mathbb{E}\left[K_t^i\right]$, which we know is negative:

$$\frac{d}{du_{t-k}^i} \mathbb{E}\left[K_{t+1}^i\right] = \left((1 - u_t^i)(1 + \gamma)(1 - P_t L_i)\right) \frac{d}{du_{t-k}^i} \mathbb{E}\left[K_t^i\right] < 0. \tag{47}$$

Thus there exists a $\lambda_e$ such that all the selfish gradients are negative, completing the proof. $\square$

### B.5 SOME SELFISH GRADIENTS ARE POSITIVE

**Theorem 3** (Self-interested mitigation). *There exists $\lambda > 0$ such that:*

$$\forall t, i, \exists k, \quad \frac{d}{du_{t-k}^i} \mathbb{E}\left[K_{t+1}^i\right] > 0. \tag{48}$$

*Proof.* Again from equation 30:

$$\frac{d}{du_t^i}\mathbb{E}\left[K_{t+1}^i\right] = -\frac{\mathbb{E}[K_{t+1}^i]}{1-u_t^i} + \mathbb{E}\left[\frac{(K_{t+1}^i)^2}{(1-X_tL_i)^2(1-u_t^i)(1+\gamma)}\sum_e\frac{\lambda_e\mu_e t}{(1+\lambda_e U_t)^2}\right]. \qquad (49)$$

We evaluate the gradient at the $U_t = 0$ policy, hence we can isolate the $\lambda_e$ where the sign flips:

$$\lambda_e^{\text{sign flip}} \times \mathbb{E}\left[\frac{(K_{t+1}^i)^2}{(1-X_tL_i)^2(1-u_t^i)(1+\gamma)}\sum_e\mu_e t\right] = \frac{\mathbb{E}[K_{t+1}^i]}{1-u_t^i}, \qquad (50)$$

which implies that at $\lambda$ sufficiently high, at least some selfish gradients become positive, and the theorem is proved. □

## B.6 MAIN RESULT

We can now state the main social dilemma theorem, which says that there is a $\lambda$ where all the individual gradients are strictly negative, but at least some social gradients are positive.

**Corollary 1** (InvestESG is a Social Dilemma). *There exists some mitigation effectiveness parameter* $\lambda > 0$ *such that*

$$\forall t, i, k, \quad \frac{d}{du_{t-k}^i}\mathbb{E}\left[K_{t+1}^i\right] < 0 \quad \text{and} \quad \forall t\,\exists i, k \quad \frac{d}{du_{t-k}^i}\sum_j\mathbb{E}\left[K_{t+1}^j\right] > 0. \qquad (51)$$

*Proof.* From the intermediate value theorem and theorems 2 and 3, we conclude that there exists $\lambda_{\text{critical}}$ for which $\exists t, i, k \frac{d}{du_{t-k}^i}\mathbb{E}\left[K_{t+1}^i\right] = 0$, and from continuity of $\frac{d}{du_{t-k}^i}\mathbb{E}\left[K_{t+1}^i\right]$ with respect to $\lambda$ we infer that there exists $\epsilon > 0$ such that for $\lambda_{\text{critical}} \geq \lambda > \lambda_{\text{critical}} - \epsilon$ we have $\forall t, i, k \frac{d}{du_{t-k}^i}\mathbb{E}\left[K_{t+1}^i\right] < 0$. Now theorem 1 applied at $\lambda_{\text{critical}}$ tells us $\exists t, i, k \quad \frac{d}{du_{t-k}^i}\mathbb{E}\left[\sum_j K_{t+1}^j\right] > 0$, and again by continuity this quantity remains positive at some smaller $\lambda_{\text{critical}} - \epsilon^*$. Thus $\lambda = \lambda_{\text{critical}} - \min(\epsilon, \epsilon^*)$ ensures that all individual gradients are negative while there exists a social gradient which is positive. □

## C  ON THE COOPERATIVE BIAS OF ADVANTAGE ALIGNMENT

Advantage Alignment has a bias towards promoting cooperation between agents when the advantages are estimated using generalized advantage estimation (GAE) (Schulman et al., 2018). The main insight is that TD-1 errors have a positive bias for on-policy trajectories because the critic lags behind the actor; advantages are systematically underestimated, producing a cooperation bias in the algorithm. The advantage alignment updated advantages are:

$$A^{*,i}(s_t, \mathbf{a}_t) = \left( A^i(s_t, \mathbf{a}_t) + \beta\gamma \cdot \sum_{j \neq i} \left( \sum_{k < t} \gamma^{t-k} A^i(s_k, \mathbf{a}_k) \right) A^j(s_t, \mathbf{a}_t) \right). \tag{7}$$

The term $\sum_{k<t} \gamma^{t-k} A^i(s_k, \mathbf{a}_k)$ acts as a multiplicative factor for the opponent advantages. If we let $b^i = \mathbb{E}_{\tau \sim \mathrm{Pr}_\mu^{\pi^i, \pi^{-i}}} \left[ \sum_{k<t} \gamma^{t-k} A^i(s_k, \mathbf{a}_k) \right]$, we can split the expression into its biased and unbiased parts:

$$A^{*,i}_{\text{biased part}}(s_t, \mathbf{a}_t) = \left( A^i(s_t, \mathbf{a}_t) + b^i \times \beta\gamma \sum_{j \neq i} A^j(s_t, \mathbf{a}_t) \right), \tag{52}$$

$$A^{*,i}_{\text{unbiased part}}(s_t, \mathbf{a}_t) = \beta\gamma \cdot \sum_{j \neq i} \left( -b^i + \sum_{k<t} \gamma^{t-k} A^i(s_k, \mathbf{a}_k) \right) A^j(s_t, \mathbf{a}_t). \tag{53}$$

It is this biased part of the new advantages which produce the cooperation bias. In the extreme case where $b \times \beta\gamma = 1$, the learning process collapses to simply doing summed-reward training.

For advantage functions $A^j(s_t, \mathbf{a})$ which perfectly evaluate the behavior of the policies $\pi_j$, the terms above have $b^i = 0$, yielding no collaborative bias for advantage alignment. However, Generalized Advantage Estimation computes the advantages as:

$$\delta_t^V := r_t + \gamma V_\phi(s_{t+1}) - V_\phi(s_t), \tag{54}$$

$$\hat{A}_t^{\text{GAE}(\lambda,\gamma)} := \sum_{l=0}^{\infty} (\gamma\lambda)^t \delta_{l+t}^V, \tag{55}$$

Where the value networks $V_\phi$ are trained to evaluate policies $\pi_\theta$ which are themselves learning and improving. As $\pi$ improves, the value networks perpetually lag behind, computing the value of slightly time-delayed policies. This implies that $V_\phi(s_t)$ will on average underestimate the next-step rewards see Kearns and Singh (2000), and this will give us a positive bias to the TD-errors seen on-policy:

$$\mathbb{E}_{\tau \sim \mathrm{Pr}_\mu^{\pi^i, \pi^{-i}}}[\delta_t^V] > 0 \tag{56}$$

$$\implies \mathbb{E}_{\tau \sim \mathrm{Pr}_\mu^{\pi^i, \pi^{-i}}} \left[ \hat{A}_t^{\text{GAE}(\lambda,\gamma)} \right] > 0 \tag{57}$$

$$\implies \hat{b}^i = \mathbb{E}_{\tau \sim \mathrm{Pr}_\mu^{\pi^i, \pi^{-i}}} \left[ \sum_{k<t} \gamma^{t-k} \hat{A}^{i,\text{GAE}(\lambda,\gamma)}(s_k, \mathbf{a}_k) \right] > 0. \tag{58}$$

This bias towards cooperation only manifests at the beginning of training, before the critic has fully caught up with the actor. At convergence of the actor, the critic does catch up, and the TD-errors become unbiased, removing the cooperative bias.

# D  EXPERIMENTAL DETAILS

Throughout our experiments, we trained using the Adam optimizer (Kingma and Ba, 2017) and trained for 70 million environment steps (approximately 20000 gradient steps), as anything beyond that resulted in lower performance. Our experiments ran in approximately 4 hours on a NVIDIA L40s gpu.

## D.1  PPO POLICY PARAMETERIZATION AND HYPER-PARAMETERS

As in the InvestESG paper (Hou et al., 2025), we use a two-layer fully-connected MLP with hidden size of 256 neurons and ReLU activations to parameterize the mean of a multivariate normal distribution. We use an additional parameter to parameterize the log standard deviation. For the critic, we use the same architecture (two-layer MLP, ReLU activations, with hidden size 256) to estimate the value directly. We swept over hyper-parameters to find the best for maximizing market total wealth. For all PPO experiments we used:

| Hyperparameter | Value |
| --- | --- |
| Algorithm | IPPO |
| Total Environment Steps | 70M |
| Number of Environments | 64 |
| Episode Length | 100 |
| Discount Factor ($\gamma$) | 0.99 |
| GAE $\lambda$ | 0.95 |
| Policy Learning Rate | 1e-4 |
| Value Function Learning Rate | 1e-4 |
| Entropy Coefficient | 0.05 |
| Clip Ratio ($\epsilon$) | 0.2 |
| Value Function Clip Range | 10 |
| Number of PPO Epochs | 4 |
| Number of Minibatches | 20 |
| Gradient Clipping | 10.0 |
| MLP Hidden Size | 64 |
| MLP Hidden Layers | 2 |
| Self-Play | False |
| Fixed Random Seed | False |

Table 1: Key PPO hyperparameters used for training in the InvestESG environment.

In the original InvestESG paper, the stochasticity of the environment is fixed with a single seed. We ran our experiments in the more challenging stochastic setting, which more accurately reflects some of the nuances of the climate problem.

## D.2 ADVANTAGE ALIGNMENT POLICY PARAMETERIZATION AND HYPER-PARAMETERS

As in theprevious section, we use a two-layer fully-connected MLP with hidden size of 256 neurons and ReLU activations to parameterize the mean of a multivariate normal distribution. We use an additional parameter to parameterize the log standard deviation. For the critic, we use the same architecture (two-layer MLP, ReLU activations, with hidden size 256) to estimate the value directly. We swept over Advantage-Alignment specific hyper-parameters (keeping the PPO hyper-parameters fixed) to find the best for maximizing market total wealth. For all Advantage Alignment experiments we used:

| Hyperparameter | Value |
|---|---|
| Algorithm | AdAlign |
| Total Environment Steps | 70M |
| Number of Environments | 64 |
| Episode Length | 100 |
| Discount Factor ($\gamma$) | 0.99 |
| GAE $\lambda$ | 0.95 |
| Policy Learning Rate | 1e-4 |
| Value Function Learning Rate | 1e-4 |
| Entropy Coefficient | 0.05 |
| Clip Ratio ($\epsilon$) | 0.2 |
| Value Function Clip Range | 10 |
| Number of PPO Epochs | 1 |
| Number of Minibatches | 20 |
| Gradient Clipping | 10.0 |
| Use RNN | False |
| MLP Hidden Layers | 2 |
| Self-Play | True |
| Advantage Alignment $\beta$ | 0.2 |
| Advantage Alignment $\gamma$ | 0.9 |
| Fixed Random Seed | False |

Table 2: Key Advantage Alignment hyperparameters used for training in the InvestESG environment.

As mentioned before, using self-play greatly improves the stability of Advantage Alignment runs across seeds, therefore we use it for all Advantage Alignment experiments.

### D.3 BASELINE SCOPE AND RATIONALE

We focus on methods that (i) handle continuous actions and high-dimensional observations, (ii) train under realistic compute budgets, and (iii) have been used in practice. This leads us to IPPO and MAPPO as the main baselines, and excludes several prior opponent–shaping (OS) methods for the full InvestESG experiments.

IPPO exposes non-stationarity and credit-assignment challenges with decentralized critics and is a strong, commonly used baseline in general-sum MARL. MAPPO follows CTDE: actors use local observations at train/test time while a centralized critic conditions on global context during training to reduce variance and improve credit assignment. In our setting, MAPPO offers a scalable, fair comparator that addresses the pathologies of IPPO without requiring execution-time communication.

| Method | Continuous actions | Practical runtime | Used at scale |
|--------|:------------------:|:-----------------:|:-------------:|
| LOLA   | ✓ | ✓ | ✗ |
| M-FOS  | ✓ | ✗ | ✗ |
| BRS    | ✓ | ✗ | ✗ |
| LOQA   | ✗ | ✓ | ✓ |
| AdAlign | ✓ | ✓ | ✓ |

LOLA requires higher-order differentiation and has been found suboptimal even on simple discrete games (see LOQA Aghajohari et al. (2024b)), and it is not designed for continuous control. M-FOS has similar limitations to LOLA with reported underperformance on small discrete settings. BRS is computationally prohibitive; reports indicate on the order of $\sim$48 A100 GPU-hours per seed on Coin Game, making InvestESG-scale sweeps impractical Aghajohari et al. (2024a). LOQA relies on normalized action probabilities, which makes it incompatible with continuous-action parameterizations. Advantage Alignment is policy-gradient based and action-agnostic, and has been deployed in higher-dimensional settings where prior OS methods are typically not compared for similar scalability reasons.

# E    ADDITIONAL FIGURES

## E.1    ABLATION OVER THE $\alpha$ PARAMETER

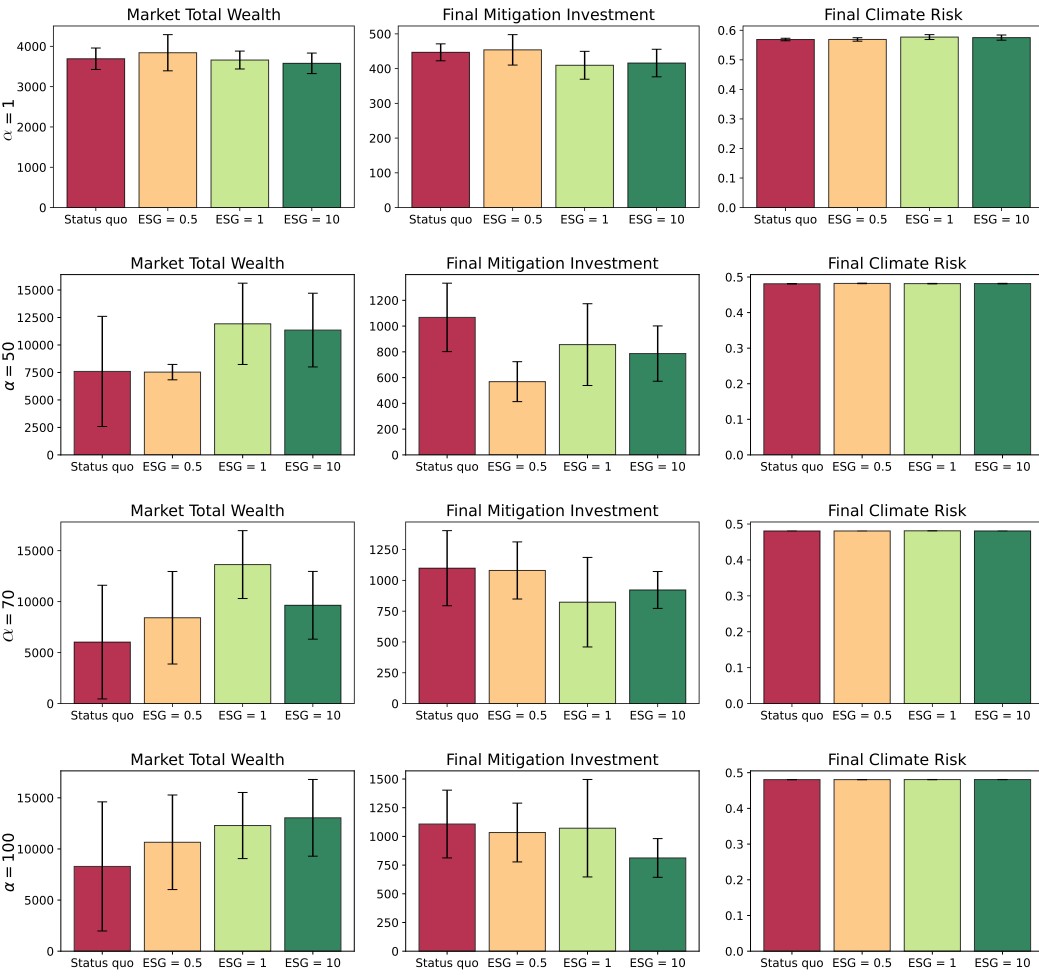

Figure 5: Comparison of final environment metrics at different $\alpha$ (introduced in equation 9) values, for 10 seeds of PPO agents. We try values of $\{1, 50, 70, 100\}$ for the parameter $\alpha$. The choice of 70 is the most sensible one, as there are clear differences between all policies with different ESG incentives. The result, albeit similar, is less apparent with a choice of $\alpha = 100$.

## E.2 ADDITIONAL RESULTS IN THE DEFAULT INVESTESG CONFIGURATION

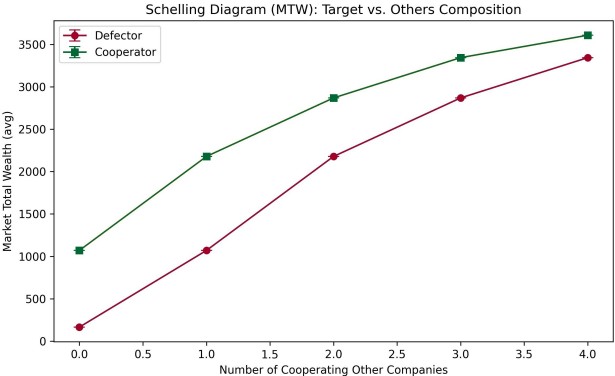

Figure 6: Schelling diagram looking at the market total wealth with a cooperator (green) or a defector (red) varying the number of other cooperating policies on the x-axis. Cooperation here is defined as in the original InvestESG paper, where a company spends $0.5\%$ of their capital on mitigation.

Figure 6 empirically shows that the cooperative action profile in the original InvestESG paper ($0.5\%$ of capital spent on mitigation) is not well calibrated. This is due to the fact that the market total wealth achieved by the cooperative action profile ($\approx 3500$) is lower than the one achieved by PPO agents ($\approx 4000$) in Figure 5. This contradicts both definition 5 and the intuitive construction that we provide of a social dilemma using *the price of anarchy* in definition 2.

### E.3 INTERPRETING NEURAL NETWORK POLICIES

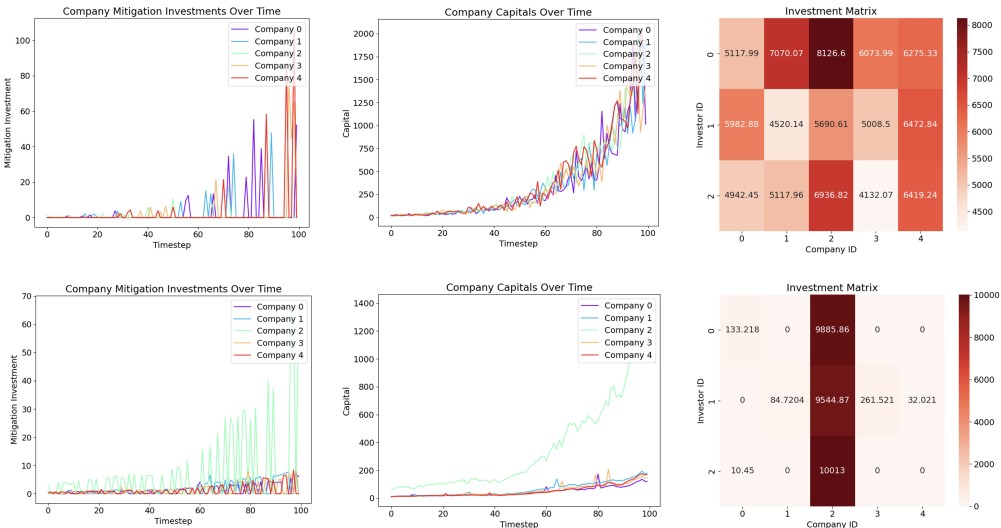

Figure 7: Policy dynamics in the ESG = 0 social dilemma. Top: Advantage Alignment (AA); bottom: PPO baseline. (i) Mitigation over time: AA invests only at climate-risk peaks and then returns to zero, whereas PPO agents invest erratically throughout. (ii) Company capital: AA keeps wealth roughly uniform across firms, while PPO gradually concentrates it. (iii) Final investment matrix: AA yields an almost symmetric allocation, whereas PPO shows fragmented, concentrated investments.

