# OpenReview forum: "Towards Sustainable Investment Policies Informed by Opponent Shaping"
_ICLR.cc/2026/Conference — ICLR 2026 Poster_

### Official Review · Reviewer_EiHm · 2025-10-24

**Soundness:** 2
**Presentation:** 3
**Contribution:** 2
**Rating:** 4
**Confidence:** 3

**Summary:**

This paper studies the InvestESG model (ICLR 2025) and provides both theoretical and empirical evidence that the Advantage Alignment Algorithm (ICLR 2025) - an opponent shaping algorithm - scales better and finds more efficient equilibria than common baselines PPO, and MAPPO. More formally, the paper equips the original InvestESG model with a parameter that it terms 'climate responsiveness parameter' and shows that a simplified version of this augmented InvestESG model exhibits characteristics of a social dilemma (unlike the general case, as claimed in the original InvestESG paper).

**Strengths:**

Originality: the paper applies an opponent shaping method to InvestESG showing that this algorithm outperforms baselines in terms of scalability and equilibrium selection. Also, if I understand correctly, the paper rectifies the claim that the original InvestESG is a social dilemma for all parameter configurations. Thus, originality comes essentially from the synthesis of two recent ideas.

Quality and Clarity: the paper is generally clearly written and rigorous with proofs being correct to the extent that I could verify. The experimental results contain enough detail to be reproducible.

Significance: the paper contributes to the literature about using ML to tackle social problems, and in this case, climate change.

**Weaknesses:**

The main weaknesses in my opinion are the following:

- While the paper is clear in its presentation, it does not discuss enough its most important modelling decisions and assumptions, i.e., whether the model of a stochastic game is indeed adequate to capture climate change and whether the climate responsiveness parameter is rich enough to capture something valuable. In particular, a value of \(\alpha=70\) seems to correspond to a rather unrealistic setting: \(\lambda\) is calibrated to reflect the 1.3 trillion commitment and scaling this by 70 seems to result in a setting of only theoretical interest. Whithout aiming to re-review the original InvestESG model, I think that this paper does not help to lift the limitations of the original model, but rather introduces some more need for discussion based on the points that I mention above.

- The paper claims as its main contribution that Advantage Alignment outperforms (various variants of) PPO in terms of equilibrium efficiency and scalability. In terms of practical importance, I am not sure what exactly we learn from this in the particular climate change setting. Shall we expect that companies will be prescribed to use the AA algorithm and, thus, achieve better outcomes in (toy model) of battling climate change? Why is opponent shaping relevant in this model and why should a company use (or a government enforce/incentivise its use). Since the paper's intented contribution seems to be the climate change mitigation, I found that the practical relevance of modeling and results was lacking.

**Questions:**

In addition to discussing the weaknesses above, I was slightly confused about the claim that the original InvestESG model is not a social dilemma for its original parameter configuration. It would be great if the authors could clarify this statement. In particular, does this rectify an inaccurate claim in the original paper where the contrary is claimed?

---

> ### Author Response · Authors · 2025-11-23
>
> We thank the reviewer for their detailed and thoughtful feedback. We address the main weaknesses below.
>
> **Response to first weakness**
>
> We agree that the value alpha = 70 is not realistic as a literal calibration of climate responsiveness. Our analysis is intended to highlight exactly this tension. What we show is that, under the default parameter values, the original InvestESG model is not a social dilemma according to our formal definition. The main point of the paper is to use mathematical modelling to prove that the climate responsiveness parameter alpha is the critical quantity that determines whether the game exhibits genuine social dilemma structure, and then to empirically identify the range of alpha for which the full InvestESG environment does become a social dilemma.
>
> The empirical threshold we find is indeed much too high to be interpreted as a realistic parameter for the real world. In our view, this should be read as a diagnosis of a limitation of the current InvestESG specification: in order for the model to be both (i) a realistic description of climate–economic dynamics and (ii) a true social dilemma, more substantial changes to the environment are needed than simply adjusting alpha. Designing and validating such a revised environment would require further domain-specific modelling work and is beyond the scope of this paper. We will clarify this perspective in the revision and explicitly position our contribution as identifying and analysing the role of alpha, rather than claiming that alpha = 70 is itself a plausible real-world setting.
>
> **Response to second weakness**
>
> We agree that the motivation for including opponent shaping can be clearer, and we appreciate the opportunity to sharpen this point. We do not expect climate companies to use Advantage Alignment (AA) as a literal decision-making procedure in a toy model of climate change. That would indeed be an unrealistic interpretation of our results. Instead, the purpose of including an opponent shaping algorithm is to obtain a more faithful behavioural model of how strategic agents adapt and respond to each other in a repeated interaction.
>
> The ultimate goal of InvestESG and related models is mechanism design: to help policy makers understand how different regulatory or financial instruments might change the behaviour of firms and investors. To do this, one must make an assumption about how agents learn and adapt. Prior work has repeatedly shown that naive RL and PPO converge to equilibria that differ from human behaviour in classic social dilemmas (for example, always-defect policies in the Iterated Prisoner’s Dilemma where humans often implement reciprocal strategies such as tit-for-tat). If we model firms as naive PPO learners, we tend to systematically overestimate how strong collaboration incentives must be in order to induce cooperative behaviour.
>
> Introducing Advantage Alignment into InvestESG is our way of correcting this modelling assumption. AA is a simple modification of PPO that explicitly accounts for the fact that other agents are themselves learning and responding, and that shapes behaviour toward cooperative yet still individually rational equilibria. Solving the environment with AA therefore yields a behavioural model that is closer to how real companies might respond to incentives than solving it with PPO alone. In practical terms, modelling economic agents with naive RL greatly overestimates the magnitude of external incentives required to achieve cooperation, while opponent shaping methods substantially reduce this gap. We include AA results in the paper precisely to illustrate the difference between a naive RL assumption about company behaviour and an opponent shaping assumption, and to argue that the latter is a more appropriate starting point for policy analysis in this type of model.
>
> **Response to question**
>
> We thank the reviewer for raising this point. Our analysis shows that, under the original parameter configuration, the default InvestESG model does not satisfy our formal definition of a social dilemma in the stochastic Markov game setting. In that sense, our work refines and qualifies the claim that InvestESG is a social dilemma "for all parameter configurations". Specifically, we identify a range of the climate responsiveness parameter alpha within which the simplified model exhibits the characteristic tension between individual and collective incentives, and show that the default parameter choice lies outside this range. We will make this clarification explicit in the revision and explain more clearly how our results relate to, and extend, the original InvestESG analysis.

---

> > ### Comment · Reviewer_EiHm · 2025-11-28
> > **Acknowledgement of rebuttal**
> >
> > Thank you for the rebuttal. I acknowledge your response and have no further questions at this stage. I will discuss the evaluation with the other reviewers.

---

### Official Review · Reviewer_TMZo · 2025-11-01

**Soundness:** 3
**Presentation:** 2
**Contribution:** 3
**Rating:** 6
**Confidence:** 3

**Summary:**

The paper applies Advantage Alignment (an opponent shaping algorithm) on a varied version of InvestESG environment, a realistic simulation of corporate and investor interactions under climate risk, and compared to baseline algorithms such as IPPO and MAPPO.
Its major contributions include:

1. Formally prove that the parameter $\alpha$ (the climate responsiveness to mitigation) is critical for making the game a social dilemma in the stochastic setting.
1. Prove AdAlign is more effective in finding social welfare maximizing strategies compared to baseline algorithms both theoretically and empirically.

**Strengths:**

1. **Clear analysis and diagnosis of InvestESG benchmark**: The paper formalizes the conditions when InvestESG is a dilemma. Then it validates the predicted 𝛼 threshold empirically: the single-firm/investor sweep exhibiting a sharp change near 𝛼≈30 and full game behavior at 𝛼=70. The proof is rigorous and the empirical implementation is well-executed. This is very useful to the community who would like to utilize this benchmark for policy analysis and policy making.
1. **Comparison between AdAlign and baselines**: In Section 5.1, the paper compares the final welfare achieved by AdAlign with the result achieved by IPPO and MAPPO, and also gives their interpretation about how opponent shaping alters investment behaviors. The analysis also include inequality analysis. In Section C.3 the paper also addresses the reason why AdAlign is the most applicable option among all the opponent shaping algorithms.

**Weaknesses:**

**Limited novelty**: The paper focuses on one *existing* benchmark (with some modifications) for deep analysis, and applies an *existing* opponent shaping algorithm on this benchmark. However, the results of original AdAlign paper show that this method has proved to be effective in maximizing social welfare compared to PPO baseline on other benchmarks. Applying the same method to a variation of InvestESG might not considered to be a *fundamental* innovation.

**Asymmetry in comparisons**: The paper claims that AdAlign benefits from self-play (one set of parameters for the company and another set of parameters for investor players) while PPO agents don't. However, it muddies conclusions about algorithmic superiority versus training-regime effects. More ablations (e.g., AdAlign without self-play; PPO with tuned self-play variants) would help.

**Questions:**

**Real-world implication**: The merit of InvestESG is rooted from its depiction of company-investor interactions in the real world. With the theoretical calculation that $\alpha$=70 leads to a full social dilemma, which makes mitigation much more potent. Can you justify this magnitude in terms of real-world ranges?

**Presentations**: The paper spends Section 2.1, 2.2, 2.3 on formulating Markov Games, RL, and Social Dilemma. The application to Markov games is relatively standard. For better presentation, the paper should focus on its novelty point.

---

> ### Author Response · Authors · 2025-11-24
>
> We thank the reviewer for their thoughtful and constructive feedback. We have incorporated the reviewer's suggestions to the presentation of the manuscript and believe that they have contributed to a cleaner and more understandable paper with its contributions dutifully highlighted. The changes have been highlighted in **green**.
>
> **Response to weaknesses:**
>
> **Limited novelty.** Our aim with this work was to make an application paper in which we establish the advantage of using Opponent Shaping as an agent modeling mechanism. We are genuinely excited by this paradigm and we believe it more closely matches the behavior of rational agents, in contrast to naive reinforcement learning. As such, we agree with the reviewer that the innovation is not “fundamental” in the sense of proposing a new algorithm. However, it is often valuable to apply techniques from one field to a new domain when this yields tangible improvements. In that regard, we think that we effectively show that it is beneficial to use Opponent Shaping in the context of climate-finance simulation. We have clarified this positioning in the revised draft at the beginning of the main theoretical and experimental sections: specifically, in the first paragraph of Section 4 (“A Formal Analysis of InvestESG as a Social Dilemma”) and in the opening paragraph of Section 5 (“Applying Opponent Shaping to InvestESG”), where the new text (marked in green in the PDF) explicitly frames our goal as demonstrating the usefulness of Advantage Alignment in this climate-finance setting.
>
> **Asymmetry in comparisons.** We agree that the presentation of this part of the paper was not clear. In the current revision we explicitly state in Section 5 (first paragraph under “Applying Opponent Shaping to InvestESG”, in green) that Advantage Alignment is trained with self-play (one set of parameters for companies and one for investors), while the best-performing PPO baselines are trained without self-play. This makes the training regime explicit rather than implicit. In addition, we will prepare a full section in the appendix reporting self-play ablations for the different algorithms. These experiments were originally conducted on a few seeds; for the camera-ready version we will rerun them with enough seeds to obtain statistically meaningful comparisons of the effect of using self-play for each method. The new appendix section will be titled *“Self-play ablations”* and will be referenced from Section 5.
>
> **Response to questions:**
>
> **Real-world implication.** We believe that the main issue with using real-world economic data to choose parameters for an environmental model is that, if the model does not match the complicated dynamics of the real world, such a calibration may have unintended consequences. InvestESG is intentionally a simple model, and in our experiments we find that with the original values of \(\lambda\) it does **not** behave as a social dilemma in the stochastic setting. In that sense, we think it is better to design models of the real world so that they satisfy key qualitative desiderata. For InvestESG, that desideratum is to behave as a social dilemma, which we see as fundamental for any climate-finance model because it captures the main reason why solving this problem is difficult in the real world: the incentives of individual agents are misaligned with social welfare. We now make this rationale explicit in the revised text at the beginning of Section 4 (the paragraph starting *“This section contains our main theoretical contribution…”*, in green), and in the \(\alpha\)-ablation discussion in Appendix A.4, where we explain that scaling \(\alpha\) is used to ensure that the environment exhibits a genuine social dilemma rather than to match a specific empirical estimate.
>
> **Presentation.** We thank the reviewer for these suggestions regarding presentation. We agree that Sections 2.1–2.3 (Markov games, RL, and social dilemmas) cover relatively standard material. To improve the presentation and focus more on the novel parts, we have shortened these subsections and move some of the more textbook-style definitions (e.g., Social Dilemma preliminaries) to the appendix, so that the main text transitions more quickly to our analysis of InvestESG (Sections 3–4) and the application of Advantage Alignment (Section 5). We hope these changes address the reviewer’s concerns about presentation.

---

### Official Review · Reviewer_b9EZ · 2025-11-01

**Soundness:** 3
**Presentation:** 3
**Contribution:** 3
**Rating:** 4
**Confidence:** 4

**Summary:**

This paper rigorously analyzes the InvestESG environment. They first formally identify the parameter $\alpha$ required to establish a true intertemporal social dilemma. The authors then apply Advantage Alignment, a scalable opponent-shaping algorithm, to this calibrated environment. This method effectively steers agents toward cooperative, high-welfare equilibria, outperforming standard MARL baselines even without external ESG incentives. The work is supported by a theoretical argument explaining why this opponent-shaping approach is inherently biased toward finding such socially beneficial outcomes.

**Strengths:**

- The paper's primary strength lies in its formal analysis of the InvestESG simulation. Instead of taking the environment at face value, the authors mathematically derive the precise conditions under which it functions as a true social dilemma.

- The successful application of Advantage Alignment to this complex high-dimensional economic simulation. It demonstrates a scalable method for finding cooperative high-welfare solutions where standard MARL baselines fail.

**Weaknesses:**

- The paper justifies excluding other OS methods like LOLA or BRS on the grounds of scalability. While reasonable, this means AA is only compared against non-shaping methods (IPPO/MAPPO). It is unclear if AA superior because it's an OS method, or because it's a better OS method. Comparing AA to at least one other OS method on a scaled-down version of the $\alpha$-InvestESG environment would be helpful to make a stronger claim.

**Questions:**

Your theoretical argument in Section 6 that AA's success stems from a "cooperative bias" induced by GAE critic lag is a central and intriguing claim. However, this mechanism is never empirically demonstrated. Could you provide data from your training runs to substantiate this? For example, could you plot the on-policy mean of the advantage estimates ($b^i$ in your derivation) over time? Seeing this term be positive and decay as the critic converges would provide strong empirical evidence that this is the active mechanism.

---

> ### Author Response · Authors · 2025-11-23
>
> We thank the reviewer for their thoughtful feedback and now address their main concerns.
>
> **Weaknesses**
>
> We agree that comparing Advantage Alignment (AA) to other opponent shaping methods on a scaled-down variant of InvestESG could, in principle, help separate the effect of using any opponent shaping at all from the effect of using AA in particular. However, our primary goal in this work is not to establish a leaderboard of opponent shaping algorithms, but to clarify how InvestESG can be used as a modelling tool for policy analysis and to argue that opponent shaping is a more realistic behavioural assumption than naive independent RL in this setting.
>
> In our view, "solving" InvestESG is only meaningful insofar as it informs mechanism design. The environment is intended to help anticipate how real-world companies and investors might respond to different climate policies, not to suggest that companies should literally run a specific RL algorithm. It is well known that naive PPO or IPPO agents behave very differently from humans in classic social dilemmas, for example converging to always-defect equilibria where humans often play tit-for-tat. Using naive PPO learners as a behavioural model therefore tends to overestimate how strong climate incentives need to be in order to induce cooperation.
>
> The purpose of introducing AA here is to provide a scalable, easy to implement opponent shaping variant of PPO that yields a more realistic model of agent dynamics than naive PPO, not to claim that AA is the uniquely best opponent shaping algorithm. This is why our main comparison axis is "opponent shaping vs no opponent shaping" rather than "AA vs all other opponent shaping methods". Appendix C.3 ("Baseline Scope and Rationale") discusses in detail why we chose AA specifically. Among existing methods, AA is, to the best of our knowledge, the only opponent shaping algorithm that both (i) scales reliably to the high dimensional, long horizon InvestESG setting, and (ii) can be implemented as a small, local modification of PPO. Our intention is that future mechanism design work on InvestESG or similar simulations can simply use AA as a default modelling assumption for learning agents, without having to engineer and tune substantially more complex opponent shaping baselines. We will clarify this scope in the revised version and explicitly defer broader claims about the relative superiority of different opponent shaping algorithms to the original Advantage Alignment paper.
>
> Concretely, if a future paper wished to study the impact of a proposed climate policy, assuming that firms are naive PPO learners would significantly overestimate the policy strength required to elicit cooperative behaviour. Our results show that replacing PPO with AA, by changing only the advantage term, already moves the predicted equilibrium behaviour much closer to cooperative, high welfare outcomes. This is the sense in which we advocate for using opponent shaping (instantiated here by AA) as the default behavioural model in InvestESG style analyses.
>
> **Questions**
>
> We appreciate the reviewer’s suggestion to empirically substantiate the "cooperative bias" mechanism described in Section 6. In the camera ready version, we will include plots showing the evolution of (i) the on policy mean of the advantage estimates and (ii) the advantage alignment term over training. As suggested, these curves illustrate that the relevant advantage term is initially positive and decays as the critic converges, providing empirical support that the GAE critic lag is indeed the mechanism driving the cooperative bias we describe.

---

### Official Review · Reviewer_mc7N · 2025-11-01

**Soundness:** 3
**Presentation:** 2
**Contribution:** 3
**Rating:** 6
**Confidence:** 3

**Summary:**

The authors develop a novel MARL-environment, Invest-ESG, that under certain parameterizations assumes the form of a social dillema. Furthermore, the authors employ Advantage Alignment (AA) for opponent shaping and compare to MAPPO and IPPO baselines. Finally, they provide a theoretical explanation of why yields superior equilibria compared to baselines.

**Strengths:**

- Originality:
The paper introduces a novel MARL environment that highlights a critical link in broader climate-economic space, namely impact investing and greenwashing risks. Furthermore, the authors make use of SOTA learning algorithms.

- Quality:

The authors are theoretically rigorous in their analysis and include an appendix proving that InvestESG is a social dillema for certain values of lambda along with ablation studies.

- Significance:

Significance largely lies in highlighting the scalability of AA even in a high-dimensional MARL context. Authors provide insight into learned policies through gini coefficient analysis, final mitigation investments, market wealth and climate risk.

Clarity:

Paper is clear and well structured.

**Weaknesses:**

-The real world impact is overstated. the problem with building international climate agreements is that cooperation is difficult to achieve, using an algorithm that is biased towards cooperative policies doesn’t really capture that phenomena. Missing some connection to actual climate-economic literature about the dynamics of impact investing.

-Theoretical results only valid under strong, unrealistic assumptions.

-Interesting components (greenwashing, resilience investments) of environment disabled.

I will reconsider my score based on the answers to the questions.

**Questions:**

Q1) Could you include results using AA without disabling greenwashing.

Q2) Have you explored utilizing heterogenous lambda values as certain industrial sectors could impact the likelihood of certain climate hazards? e.g. extractive industries and manufacturing.

Q3) Are there existing econometric models of impact investing that are to some degree comparable with the results seen here?

---

> ### Author Response · Authors · 2025-11-23
>
> We thank the reviewer for their detailed and considerate feedback and now address the main concerns.
>
> **Response to weaknesses**
>
> We agree that the direct impact of this paper on near term policy making is limited. Our primary objective is to help establish foundations for how artificial intelligence, and in particular MARL, can eventually be used in a principled way for climate related policy analysis. The main contribution is therefore a problem formalization and modelling contribution, which required some technical work but is not itself a full policy instrument.
>
> Our focus was on setting the right inductive biases so that (i) the problem remains solvable by RL agents, while (ii) still capturing the core strategic tension of climate negotiations. First, we modified the simulator so that it actually exhibits the misalignment between individual and collective incentives that one expects in international climate agreements, and we did this under a formal social dilemma framework for Markov games. Second, we argued for a change in the default learning paradigm: like humans, RL agents should be aware that other agents are themselves adapting, so we advocate using opponent shaping methods rather than standard naive RL. Finally, we showed empirically that this shift in modelling assumption leads to substantially different, and more cooperative, equilibrium policies in the calibrated social dilemma regime.
>
> Regarding the concern that AA is “biased” toward cooperative policies and that this may not reflect the difficulty of real world cooperation, we agree that this would be problematic if AA were simply hard coding altruistic behaviour. However, the key point is that Advantage Alignment is proven to converge to Nash equilibria under its assumptions, so the cooperative outcomes it finds are still individually rational best responses. In other words, AA does not force agents to sacrifice their own returns for the common good; instead, it changes the learning dynamics so that agents discover equilibria that are both cooperative and stable, when such equilibria exist. This is why we view AA as a useful behavioural model rather than an unrealistic “cooperation oracle”.
>
> The criticism that the theoretical results rely on strong and somewhat unrealistic assumptions is also valid. These assumptions were chosen deliberately to make the analysis tractable and to isolate the mechanism that turns the simplified game into a social dilemma. We see the theory as providing conceptual guidance and motivation for the empirical modifications to InvestESG, rather than as a literal description of the full climate economic system.
>
> Finally, it is a fair point that some interesting features of InvestESG, such as greenwashing and resilience investments, were disabled in our main experiments. For the camera ready version we will include ablations with greenwashing and resilience enabled.
>
> **Responses to questions**
>
> Q1) Could you include results using AA without disabling greenwashing?
>
> Yes. For the camera ready version we will include AA results with greenwashing enabled (and, where space permits, with resilience investments enabled as well).
>
> Q2) Have you explored utilizing heterogeneous lambda values?
>
> No. In this work we focus on a homogeneous lambda in order to keep the parameter space manageable and to clearly analyse the baseline social dilemma structure.
>
> Q3) Are there existing econometric models of impact investing that are to some degree comparable?
>
> Yes, there are econometric and equilibrium models of impact investing and climate finance, but these typically assume fixed behavioural rules or rational expectations rather than learning agents. Our work is complementary to that literature, providing an agent based, learning centric perspective on how adaptive firms and investors might respond to incentives in an impact investing setting.

---

### Meta-Review · Area_Chair_RBL2 · 2026-01-16

**Summary:**

The paper investigates the use of the Advantage Alignment opponent-shaping algorithm within the InvestESG simulation to address the social dilemmas inherent in climate change investment policies. The authors formally characterize the conditions under which the simulation functions as an intertemporal social dilemma, identifying the climate responsiveness parameter as a critical threshold where individual incentives diverge from collective welfare. Empirical results demonstrate that Advantage Alignment outperforms standard reinforcement learning baselines, such as IPPO and MAPPO, by effectively influencing agent learning dynamics toward more cooperative and sustainable outcomes.

**Reviewer Concerns:**

There were a few concerns about the empirical validation of this work. The theoretical claims were appreciated. It seems that the authors addressed some of the concerns in the rebuttal.

**Reviewer Scores:**

It seems like a borderline paper and the AC recommends acceptance with reservations. Unfortunately, he cannot forsee if the negative reviewer would have changed their scores after the rebuttal.

---

### Decision · Program_Chairs · 2026-01-26

Accept (Poster)